# Topical Steroids and Glaucoma Filtration Surgery Outcomes: An In Vivo Confocal Study of the Conjunctiva

**DOI:** 10.3390/jcm11143959

**Published:** 2022-07-07

**Authors:** Leonardo Mastropasqua, Lorenza Brescia, Francesca D’Arcangelo, Mario Nubile, Giada D’Onofrio, Michele Totta, Fabiana Perna, Raffaella Aloia, Luca Agnifili

**Affiliations:** 1Ophthalmology Clinic, Department of Medicine and Aging Science, University G. d’Annunzio of Chieti-Pescara, 66100 Chieti, Italy; mastropa@unich.it (L.M.); francescadarcangelo@gmail.com (F.D.); m.nubile@unich.it (M.N.); giada01.88@hotmail.it (G.D.); michetto135@gmail.com (M.T.); f.perna@iapb.it (F.P.); raffaellaaloia77@gmail.com (R.A.); l.agnifili@unich.it (L.A.); 2International Agency of Prevention of Blindness, 00185 Rome, Italy

**Keywords:** glaucoma filtration surgery, surgery outcome, steroids, conjunctiva, ocular surface preparation, in vivo confocal microscopy

## Abstract

(1) Background: The purpose of this study is to investigate the effects of topical steroids on conjunctiva in patients undergoing filtration surgery (FS) for glaucoma by using confocal microscopy (CM); (2) Methods: One hundred and four glaucomatous patients were randomized to fluorometholone or lubricants four weeks before FS. CM was performed before treatments and pre-operatively. Dendritic and goblet cell densities (DCD, GCD), stromal meshwork reflectivity (SMR), vascular tortuosity (VT), and intra-ocular pressure (IOP) were the main outcomes. By evaluating treatments and outcomes (12-month success/failure) as categorical variables, patients were grouped into Group 1, 2, 3, or 4 (success/failure with fluorometholone, or lubricants); (3) Results: Twelve-month IOP was reduced in Groups 1 and 3 (*p* < 0.001). After treatments, DCD and SMR were reduced in Groups 1 and 2 (*p* < 0.01), and 1 and 3 (*p* < 0.05), respectively. Pre-operative DCD was lower in the steroid compared to lubricant group (*p* < 0.001), whereas SMR was lower in successful (1 and 3) compared to failed groups (2 and 4) (*p* = 0.004). There were no significant differences between the fluorometholone and lubricant groups for success percentages. The number of bleb management procedures and IOP lowering medications were lower in Group 1 compared to Groups 2–4 (*p* < 0.05); (4) Conclusions: Topical steroids mitigate conjunctival inflammation and lower the stromal density in patients undergoing FS. These modifications lead to less intensive post-operative management.

## 1. Introduction

More than fifty years after its introduction, filtration surgery (FS) still represents the most diffuse surgical procedure to reduce the intra-ocular pressure (IOP) in patients with medically uncontrolled glaucoma. FS works by creating an intra-scleral fistula that drains aqueous humor (AH) from the anterior chamber toward the sub-conjunctival space, thus forming a resorption structure known as a filtration bleb [1,2,3]. However, the filtration bleb may fail in its function over time for different reasons.

As widely demonstrated, glaucoma therapy-related conjunctival modifications may significantly affect the bleb filtration capabilities after FS, since IOP lowering medications damage, disturb, or stimulate components that are involved in the AH outflow through the bleb-wall [4,5,6,7,8,9,10,11]. These components include goblet cells (GCs), stromal collagen bundles, inflammatory dendritic cells (DCs), and blood or lymphatic vessels. In detail, GCs work as carriers of the AH, whereas stromal collagen hinders AH resorption by increasing the bleb-wall resistivity to fluids outflow. On the other hand, DCs and blood vessels stimulate inflammation, affecting the AH carriers and stimulating collagen deposition [3,8,9,11]. The progressive alterations of these structures increase the risk of inadequate AH drainage and surgical failure [4,5,6,7,12]. Therefore, the need to manage the ocular surface before FS, especially by mitigating inflammation, is keenly felt by surgeons [13,14].

To date, very few studies have investigated whether pre-operative strategies aimed at mitigating ocular surface disease (OSD) can favorably affect the FS outcome. In a randomized placebo-controlled trial, the use of topical ketorolac or fluorometholone one month before trabeculectomy reduced the likelihood of post-operative bleb needling or the need for IOP lowering drugs, with better results in patients receiving steroids compared to those taking ketorolac [15]. In another study, Lorenz et al. found that a simplified pre-operative medication regimen, which comprised only a preservative-free (PF) fixed combination of dorzolamide/timolol, reduced the ocular surface inflammation and was as effective as topical steroids and systemic acetazolamide in terms of surgical outcomes [16]. Nevertheless, though the reduction of pre-operative conjunctival inflammation is believed to limit scarring within the bleb-wall after FS, studies evaluating the conjunctival changes induced by steroids at the site of FS at a cellular level are still lacking.

In vivo confocal microscopy (IVCM) has been widely used to describe the conjunctiva in patients with glaucoma, either during the medical management of the disease or in the pre-operative period [7,8,9,11]. In fact, IVCM is able to evaluate the above-mentioned components involved in the AH outflow after surgery, which can be considered as potential indicators of surgical outcomes. This is crucial, because a pre-operative assessment may guide the clinician in adopting personalized peri-operative strategies.

The aims of the present study were to evaluate, using IVCM, the impact of topical steroids on conjunctival GCs, DCs, stromal density, and blood vessels on FS outcomes, and to evaluate whether confocal features could play a role in predicting success or failure in patients undergoing FS.

## 2. Materials and Methods

### 2.1. Study Population

This prospective, placebo-controlled, single center study adhered to the tenets of the Declaration of Helsinki and was approved by the Institutional Review Board of Department of Medicine and Aging Sciences, University G. d’Annunzio of Chieti and Pescara, Italy. All eligible patients who agreed to participate in the study signed an informed consent form before enrollment, after receiving an explanation of the nature and possible consequences of the study. Consecutive patients scheduled for first-time FS (trabeculectomy) were enrolled at the Ophthalmology Clinic, National High-Tech Eye Center (NHEC) of the University G. d’Annunzio of Chieti and Pescara, Chieti, Italy.

Inclusion criteria were: open angle glaucoma with uncontrolled IOP under maximal tolerated medical therapy (three active compounds, comprising the short-term use of systemic oral acetazolamide when required); progression of visual field (VF; Humphrey Field Analyzer (HFA), Guided Progression Analysis software) damage, as confirmed on three consecutive reliable tests; and IOP lowering therapy regimen unmodified in the last three months. Exclusion criteria were: secondary glaucoma; history of concomitant intra-ocular or ocular surface disease that could potentially contraindicate the use of topical steroids; known allergy to steroids; bulbar trauma; systemic or topical therapies in the last 6 months, potentially affecting the ocular surface; any previous ocular surgery (excluding cataract); use of contact lenses; and pregnancy. If both eyes were eligible for the study, only the eye with the more advanced perimetric stage was included.

### 2.2. Pre-Operative Treatments and Peri-Operative Considerations

After enrollment, patients were randomized (by a computer-based randomization program in a double-blinded fashion) to receive fluorometholone 0.1% (Fluaton, Bausch & Lomb-IOM SpA, Aci Sant’Antonio, Catania, Italy; Group A) or lubricants (Hyalistil BIO, SIFI, Aci Sant’Antonio, Catania, Italy; Group B). All study medications contained benzalkonium chloride (BAK) as a preservative. The labels of the study medication bottles were concealed from the patient as well as the physician.

In the 4 weeks preceding the date of surgery, patients were instructed to administer in the eye scheduled for FS one drop of their assigned medication four times daily, and to continue unmodified their IOP-lowering medical therapy. Two weeks after the initiation of pre-operative treatment, a safety check with IOP measurement was scheduled.

All patients underwent a 0.02% mitomycin-C (3 min, in soaked sponges) augmented trabeculectomy, as previously described [17]. All surgical procedures were performed using a standardized technique, which required the creation of a fornix-based conjunctival flap and a 4 × 4 mm, 300-µm thick scleral flap, which was sutured down with four 10-0 nylon sutures. To limit post-operative inflammation, the conjunctival flap was sutured down using 10-0 nylon sutures.

As per protocol, the post-surgical therapy required the use of PF dexamethasone eye drops four times daily (tapered in 12 weeks) and PF levofloxacin four times daily (discontinued after 2 weeks).

Patients were examined weekly during the first month; then, the frequency of controls was scheduled in accordance with the post-operative course, considering as clinical indicators the filtration bleb features at slit lamp (Mainz Bleb Appearance Grading System) and IOP [18]. The last follow-up was planned at 12 months.

Each decision on the timing for additional procedures was made according to post-operative IOP and bleb morphology. If IOP progressively increased and was not considered at target for the patient [19], and bleb features tended to become encapsulated or flat, management strategies including laser suture lysis, bleb needling with 5-fluorouracil, or bleb revision with mitomycin-C were sequentially adopted. When all management strategies failed to control IOP, anti-glaucoma therapy was restarted.

At the 12-month visit, surgery was considered successful when baseline IOP was reduced by at least 30%, with or without anti-glaucoma medications; otherwise, it was considered to have failed. In case of failure, patients abandoned the study and received additional glaucoma surgery.

### 2.3. Examinations

At baseline, before initiating treatments, patients underwent a complete examination, with the determination of the best corrected visual acuity, slit lamp ocular surface and lid assessment, IOP measurement (Goldmann applanation tonometry, AT900^®^, Haag Streit Diagnostics, Koeniz, Switzerland), anterior and posterior segment evaluation, and VF test (HFA 24-2 test, SITA-standard). Twenty-four hours later, patients underwent IVCM of the conjunctiva, which was repeated at the same site after four weeks, i.e., the day before the FS.

### 2.4. IVCM of the Bulbar Conjunctiva

IVCM (HRT III Rostock Cornea Module, diode-laser 670 nm; Heidelberg Engineering, Heidelberg, Germany) was performed at the upper bulbar conjunctiva, with particular attention at the site corresponding to the bleb formation after FS, to evaluate the structures involved in AH resorption, that is, epithelial GCs and DCs, stroma, and blood vessels. GCs and stroma can be considered as pre-surgical indicators of further trans bleb-wall AH flow and resistivity, respectively. DCs and blood vessels can be considered as indicators of glaucoma therapy-related ocular surface inflammation and can be used to estimate the post-surgical fibrotic reaction of the conjunctiva [9]. Two masked investigators (LB, investigator 1; MT, investigator 2) independently calculated the GCD and DCD to assess the interobserver variability.

### 2.5. Confocal Parameters

(i)Goblet cell density (GCD): in accordance with the definition and images provided in previous confocal studies, GCs had to appear as oval-shaped and hyperreflective cells, dispersed within the epithelium (or crowded in groups), larger than the surrounding epithelial cells, and located at a depth of 10–30 µm [7,8,9,11]. As documented, GCs act as carriers of the AH through the bleb-wall epithelium after filtration surgery [8,9,11]. The cell count software (Heidelberg Engineering GmbH) of the confocal microscope was used to determine the GCD (cells/mm^2^ ± SD) in manual mode.(ii)Dendritic cell density (DCD): the definition and features of DCs had to be consistent with those reported in literature [9,11,20]. DCs can appear mature and activated (hyper-reflective and elongated body, with membrane processes and frequently crowded in clusters) or immature and silent (large body with rare membrane processes, if any). They are located within the epithelium and the basal membrane of the conjunctiva (10–30 µm to 30–50 µm of depth) and act as peripheral effectors of the immune system, stimulating inflammation and fibrosis in response to toxic stimuli [20,21]. As for GCD, the cell count software was used to determine the DCD (cells/mm2 ± SD) in manual mode.(iii)Stromal meshwork reflectivity (SMR): as for GCs and DCs, the characteristics of the conjunctival stroma at IVCM had to be consistent with those reported in previous studies [9,22]. This layer (50–150 µm of depth) appears loosely arranged with thin collagen fibers and some blood vessels in the external portion but presents a denser, fibrous network with bundles of collagen fibers—occasionally hosting blood vessels, cystic spaces, and inflammatory cells—in the internal portion [22,23,24]. The tissue reflectivity represents an indirect confocal indicator of the amount of collagen fibers contained within the stroma and, thus, is a surrogate measure of the conjunctival resistivity to fluid movement. SMR was calculated as previously described, i.e., by determining the average gray value of a selected high-quality image, using the Image J software (http://imagej.nih.gov/ij/ accessed on 21 September 2021; provided in the public domain by the National Institutes of Health, Bethesda, MD, USA) [9]. SMR was graded as follows: normal (Grade 0: average gray value <90), mild (Grade 1: gray value between 90.01 and 105), moderate (Grade 2: gray value between 105.01 and 125), and high (Grade 3: >125.01), reflectivity; according to this grading scale, grades 0 to 3 corresponded to a loosely, mildly, densely, and very densely arranged stromal networks, respectively.(iv)Vessel tortuosity (VT): conjunctival vessels are located in the sub-epithelium or superficial stroma, and less frequently in the deep fibrotic portion of the stroma. They appear as broad black linear or slightly curved structures with parallel sides, frequently showing hyper-reflective and round-shaped cells within the lumen [22,23,24]. VT was assessed according to a previously adopted grading scale with four grades: straight (0), mild (1), moderate (2), and severe (3) [22]. Vessel tortuosity is an indirect indicator of stromal fibrosis and a direct indicator of chronic local inflammation [25].

### 2.6. Confocal Microscopy Procedure

The details of the confocal microscopy procedure to analyze the upper bulbar conjunctiva in patients undergoing FS have been previously described [9]. Briefly, with the patient instructed to direct the gaze downward, the entire upper bulbar conjunctiva (superior nasal, superior central, and superior temporal portions) was analyzed.

Images were acquired at the epithelium (10–30 µm of depth) to evaluate GCs, at the sub-epithelium (10–30 µm) to evaluate DCs, and from the stroma (50–150 µm) to evaluate the SMR. The automatic brightness mode was selected during all examinations.

Sixty images (20 images per each portion; field of view of 400 × 400 µm) were acquired in both the pre-and post-treatment sessions; among all the acquired images, 15 randomly selected, high-quality scans per case (five per portion) without motion blur or compression lines were selected for the analysis. Two different, experienced IVCM operators performed the confocal examinations and selected the images (LB and MT); a second experienced operator evaluated the images (GDO). All the operators were masked regarding the patient history and group assignment.

### 2.7. Outcomes of the Study

Pre- and post-treatment sessions, and 12-month data were considered the time points for the statistical analyses. The main outcome measures were the modification of GCD, DCD, SMR, and VT, and the success or failure at 12 months based on the relative IOP reduction as compared with baseline IOP. Differences among treatments for post-operative bleb management procedures (laser suture lysis, needling, needling revision) and IOP lowering medications, and correlations between post-surgical procedures, IOP lowering medications, and 12-months IOP and group classification were also evaluated.

### 2.8. Statistical Analysis

The sample size was determined based on a success rate of 100% in the group treated with steroids (Group 1) and 76% in the control group (Group 2) [15]. The alpha level was set to 0.05 and the power to 80%. To minimize the number of patients with the worse outcome, the enrollment ratio was set to 0.6, i.e., 3 out of every 5 patients were assigned to the group that would receive steroid therapy prior to surgery. The required sample size was determined to be 53 patients: 33 in the steroid group and 20 in the non-treated group. Enrolled patients were assigned to treatment groups using a block (size = 5) design.

The two treatment groups (A, steroid and B, lubricant) and the two possible outcomes (success or failure at 12 months) were evaluated as categorical variables and further grouped in the following manner: Group 1 and 2, success or failure after steroid treatment; Group 3 and 4, success or failure without steroid treatment. Patient characteristics for the treatment groups were evaluated using chi squared and student’s *t*-test, as appropriate. Variations from baseline to pre-operative values of IOP, DCD, GCD, and SMR were evaluated using an ANOVA with post-hoc Tukey HSD test, and VT with a Kruskal-Wallis Test.

Moreover, the canonical discriminant analysis (CDA) was used to discern the four groups using the percentage variation between baseline and pre-operative clinical and confocal microscopy parameters, and to estimate the impact of steroid treatment on the surgical outcomes. Statistical significance for backward selection was determined using an α = 0.05. Leave-one-out cross-validation was used to validate the predictive model.

Spearman’s non-parametric coefficient was used to investigate the correlations between post-surgical procedures, IOP lowering medications, and 12-month IOP and group classification.

Statistical analyses were performed using SPSS software (version 26, International Business Machines Corp., Armonk, NY, USA). MedCalc Statistical Software version 16 (MedCalc Software Ltd., Ostend, Belgium) was used to determine sample size.

## 3. Results

### 3.1. Clinical Results

All patients completed surgery without complications, and none was lost to follow-up. Table 1 reports the demographic and clinical characteristics of the four groups. Pre-operative IOP did not significantly change compared to baseline in any of the groups and did not differ among groups at both time points. Significant differences were found between baseline and 12 months IOP for Groups 1 and 3 (*p* < 0.001).

There were no significant differences between groups treated with fluorometholone (1 and 2) and control groups (3 and 4) with regard to the percentage (%) of patients with complete success (*p* = 0.08) or qualified success (*p* = 0.37). Conversely, the numbers of post-operative bleb management procedures and IOP lowering medications were lower in Group 1 compared to Groups 2–4 (*p* < 0.05). Figure 1 shows the jittered distribution of overall bleb management procedures among groups.

### 3.2. IVCM Results

The four treatment groups did not present statistically significant differences in baseline DCD, GCD, SMR, and VT, or in pre-operative, GCD, and VT. Compared to baseline, DCD was significantly reduced in Groups 1 and 2 (*p* < 0.001 and *p* < 0.01, respectively), whereas SMR was slightly reduced in Groups 1 and 3 (*p* < 0.05). Pre-operative DCD was significantly lower in Groups 1 and 2 compared to control groups (*p* < 0.001), whereas SMR was significantly lower in successful (1 and 3) compared to failed groups (2 and 4) (*p* = 0.004). VT and GCD did not change between baseline and pre-operative time points in any of the groups, but pre-operative GCD was significantly higher in Groups 1–3 compared to Group 4 (*p* < 0.001) (Table 2). GCD (28.72 ± 2.14 and 28.91 ± 3.21, respectively), and DCD (58.22 ± 7.32 and 58.97 ± 6.99, respectively) estimated by investigator 1 (LB) were not significantly different (*p* > 0.05) from that estimated by the investigator 2 (MT).

### 3.3. Canonical Discriminant Analysis

CDA was used to distinguish groups using two discriminant functions, based on DCD and SMR, which were the highly significative variables found between baseline and pre-operative time points. Specifically, the standardized CDA was based on the percentage variation of DCD and SMR, where the coefficients were 0.957 and 0.334 for function 1 and −0.294 and 0.943 for function 2 (Figure 2). Both functions were able to discriminate the four groups (Wilks’s lambda < 0.001), while each function alone did not perform as well (Wilks’s lambda = 0.001). Functions 1 and 2 correctly classified 90.6% of the original grouped patients. Leave-one-out cross-validation, where each case was classified by the functions derived from all other cases, correctly classified 86.8% of cases, thus validating the results of predictive correlation between the modification of DCD and SMR in the 4 weeks before surgery and the success/failure of trabeculectomy at 12 months. The number of post-surgical procedures and the number of post-operative IOP lowering medications positively correlated with 12-month IOP (*p* < 0.001, rho = 0.406) and group classifications (*p* < 0.001, rho = 0.518). Figure 3 is a mosaic of confocal frames taken from two representative patients, showing the modification of the confocal parameters between baseline and the pre-operative time point in successful (Group 1) vs. failed cases (Group 4).

## 4. Discussion

The bulk of the literature clearly demonstrates that the pre-operative features of the ocular surface and the outcomes of glaucoma filtration surgery have a strong mutual relationship [6,14,26]. Therefore, improving the ocular surface quality plays a critical role in the management of patients who are candidates for surgery [13,14].

In the present study, IVCM was, for the first time, used to evaluate the effects of topical steroids on the most crucial conjunctival structures involved in the AH resorption through the bleb-wall. In detail, we found that unpreserved fluorometholone reduced pre-operative DCD by 40% and 30%, in successful and failed surgeries (Groups 1 and 2), respectively. Moreover, fluorometholone slightly reduced (9%) SMR only in patients that underwent a successful FS (Group 1). Interestingly, these modifications were not associated with an increase in the success rate of surgery, but showed a significantly reduced need for post-operative bleb manipulation procedures or the use of IOP lowering eyedrops.

These results are in agreement with previous findings, highlighting that the ocular surface improvement induced by low-potency steroids such as fluorometholone was associated with less intensive post-operative management [15,27]. The novelty of our study lies in the fact that we investigated, in vivo and at a microscopic level, the cellular modifications underlying the clinical effects of topical steroids.

DCs represent a key cellular population in patients with glaucoma; their activation and proliferation were shown to contribute to ocular surface inflammation and to increase the risk of bleb dysfunction over time [7,9,20]. Similarly, the presence of a pre-operative, dense conjunctival stroma was found to exert a negative influence on the surgery outcome [9]. Therefore, the pre-operative reduction of DCD and SMR induced by topical steroids may positively affect bleb functionality after surgery by mitigating inflammation and limiting tissue fibrosis.

These results agree with the findings of Baudouin et al., who observed a significant reduction of a key inflammatory marker, human leukocyte antigen (HLA)-DR, after one month of therapy with fluorometholone, and with the observation that steroids have significant anti-proliferative activity on Tenon’s fibroblast cell line cultures, reducing collagen deposition [27,28].

As stated above, fluorometholone-related DCD and SMR reductions led to less intensive post-operative management. Specifically, this was documented only in patients who had the greatest inflammation reduction and a partial improvement of the stromal density of the conjunctiva (Group 1). In fact, Group 1 presented a 10% higher DCD reduction compared to Group 2, with a decrease in SMR of almost 10% (SMR did not change in Group 2). On this basis, one may hypothesize that steroid treatment may influence surgery outcomes only if the inflammation mitigation surpasses a certain threshold (at least one third reduction in the tested parameters) and when the inflammation mitigation is coupled with a slight decrease in tissue fibrosis. These considerations appear to be in line with a pioneer study in which Broadway and co-workers found that the pre-operative use of fluorometholone 1% increased success rates by reducing both the density of inflammatory cells and fibroblasts within the conjunctiva [29].

A critical interpretation of these observations is that the inflammation mitigation induced by steroids, when under a certain limit, is only useful to reduce the need for post-operative management, rather than to increase the likelihood of surgical success [15]. It is likely that in order to achieve an increase in the success rate of FS, the reduction of pre-operative inflammation and of stromal density should be greater than 40% and 10%, respectively. This could be obtained with longer duration treatments or by using more potent steroids.

The results observed in Group 3 raise different considerations. This group that received topical lubricants had the same percentage of successful surgeries compared to Group 1, without DCD reduction and with mild SMR improvement. On the other hand, Group 3 required a higher number of bleb manipulation procedures and post-operative IOP lowering medications compared to Group 1 (3 times more).

Based on these findings, one cannot completely rule out a potential impact of sodium hyaluronate on surgical outcomes. As is known, the use of sodium hyaluronate represents one potential strategy to prepare patients for FS, since it mitigates dry eye and, thus, may contribute to reducing inflammation [13,14]. In addition, sodium hyaluronate seems to stimulate GCs, which play an active role in the aqueous humor resorption through the bleb-wall after FS [8,9,11]. Nonetheless, in our study, GCD did not increase with the use of lubricants, probably because of the short duration of treatment.

Finally, as an ancillary part of this study, we also evaluated whether IVCM could be useful in providing cellular predictive biomarkers of surgical outcome in patients receiving pre-operative management with steroids.

As expected, because of their importance in bleb functionality, the discriminant analysis found that DCD and SMR, only when considered together, were highly significative confocal variables which correctly distinguished groups in 90% of cases. In this way, CDA further confirmed the impact of steroids on FS outcomes, since the analysis showed that patients presenting a higher reduction of pre-operative DCD and SMR were part of Groups 1 and 2.

The present study presents some limitations. First, we did not consider patients treated with topical non-steroidal anti-inflammatory drugs (NSAIDs). However, previous studies documented that the effects of steroids and NSAIDs on FS outcomes are similar, although a significantly higher efficacy of steroids over NSAIDs was described [14,27,28]. Second, since SMR is an arbitrary index and an indirect indicator of the tissue density, this parameter could not clearly indicate the collagen amount of the conjunctival stroma. Third, because we did not consider intermediate follow-up time points between pre-operative and 12 months, we are unable to describe how conjunctival features progressively changed during the first year after surgery. Fourth, we did not investigate the relations between IVCM variables and post-operative bleb features. This may be an additional important factor that could further clarify the impact of pre-operative steroids on surgical outcomes. Finally, we did not repeat IVCM at last follow-up to evaluate whether confocal parameters and 12-month follow up were correlated. However, based on a previous confocal study which found that the bleb-wall features were better when the ocular surface was less inflamed, we can hypothesize that steroid-treated patients may develop a more favorable bleb morphology after surgery [30]. Further studies aimed at investigating whether GCD, DCD, SMR, and surgical success rate correlate are warranted to clarify this aspect.

Furthermore, considering the potential effects of sodium hyaluronate on AH resorption through the bleb-wall, a prospective study aimed at investigating the impact of this treatment on FS outcomes is warranted.

## 5. Conclusions

In conclusion, the present study confirmed the positive effects of topical, low-potency steroids in promoting less intensive post-operative management after FS, rather than in increasing the success rate of surgery. At the same time, this study unraveled, in vivo, the cellular modifications underlying the mechanism of action of steroids, and preliminarily found IVCM to be a potentially useful system to predict surgical outcomes.

## Figures and Tables

**Figure 1 jcm-11-03959-f001:**
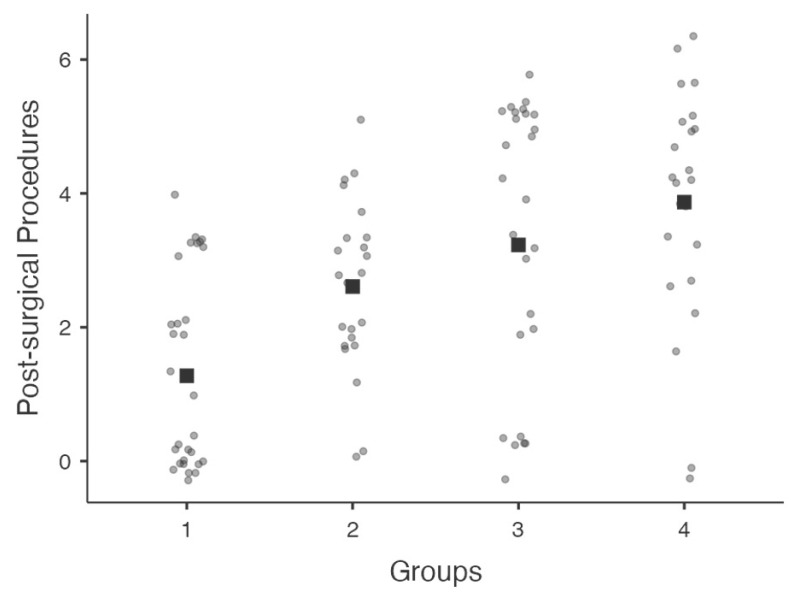
Graphical distribution of jittered data (values are plotted as dots along each axis) among groups representing post-surgical bleb management procedures. Median value is representing by squares along axes.

**Figure 2 jcm-11-03959-f002:**
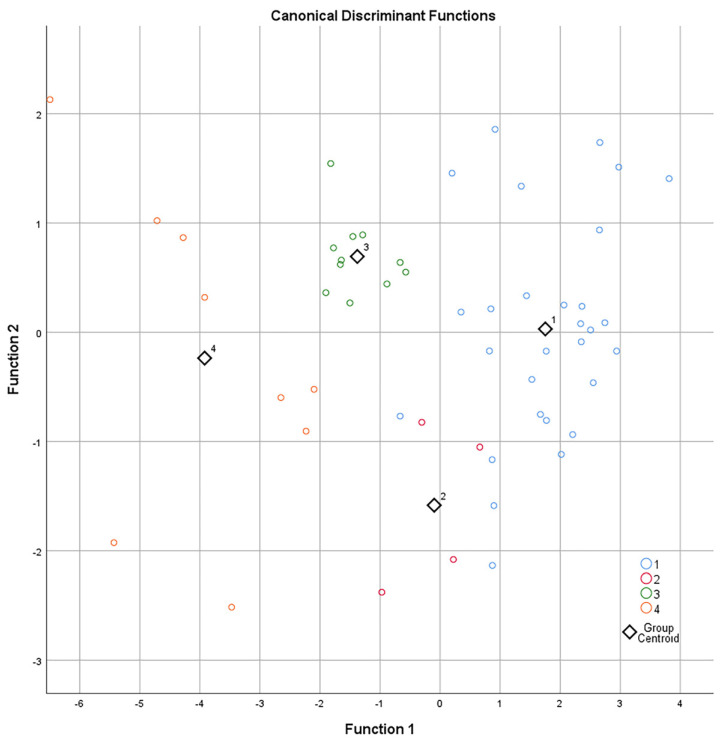
Scatter plot generated with the two functions obtained with the canonical discriminant analysis of patients belonging to the four treatment-outcome groups. Function 1: +0.957 (delta DCD) and +0.334 (delta SMR). Function 2: −0.294 (delta DCD) and +0.943 (delta SMR). These functions correctly classified 90.6% of the original grouped patients and validation with leave-one-out cross-validation correctly classified 86.8%.

**Figure 3 jcm-11-03959-f003:**
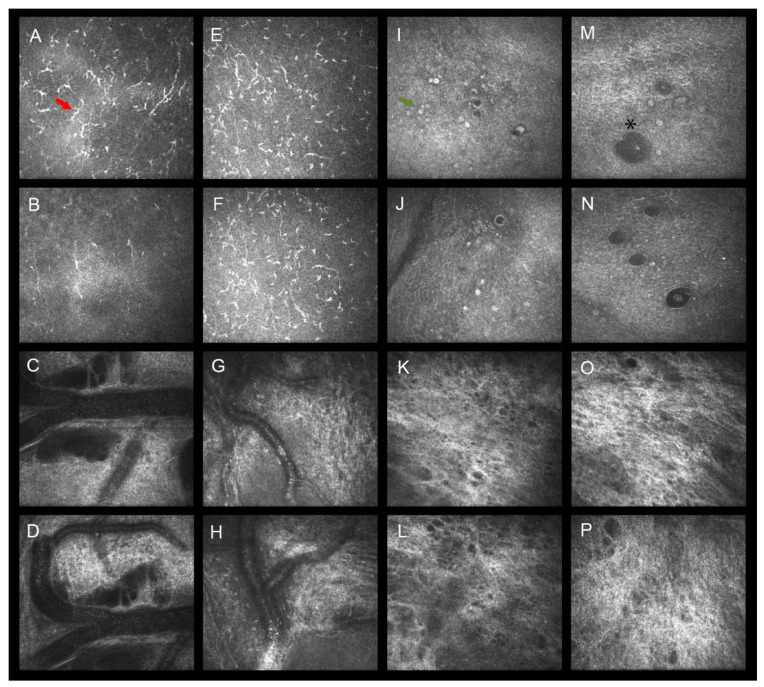
Baseline and post-treatments confocal frames of the conjunctival epithelium and stroma in representative patients that received topical fluorometholone (**A**–**D**,**I**–**L**; Group 1) or lubricants (**E**–**H**,**M**–**P**, Group 4). Compared to baseline (**A**), fluorometholone significantly reduced sub-epithelial DCs (red arrow) four weeks after treatments (**B**), whereas lubricants did not (**E**,**F**). Epithelial GCs (green arrow) did not change their baseline density neither in patients receiving steroids (**I**,**J**) nor in controls (**M**,**N**) with an intra-epithelial conjunctival microcyst surrounded by goblet cells recognizable (asterisk, **M**). The baseline vessel tortuosity did not change after steroid (**C**,**D**) or lubricant (**G**,**H**) treatments. The stromal meshwork reflectivity presented a mild reduction compared to baseline in patients receiving fluorometholone (**K**,**L**) while this was not observed in patients receiving lubricants (**O**,**P**).

**Table 1 jcm-11-03959-t001:** Demographic and clinical data.

	Gender (M/F)	Age (Years ± SD)	Time on Therapy (Months ± SD)	Baseline IOP (mmHg ± SD)	Pre-OP IOP (mmHg ± SD)	12 Months IOP (mmHg ± SD)	Post-OP Procedures (*n* ± SD)	Post-OP IOP Lowering Medications (*n* ± SD)
Group 1	13/16	68.34 ± 7.97	72.5 ± 3.2	29.55 ± 5.32	28.24 ± 4.57	15.51 ± 2.28 ^†‡^	1.28 ± 1.01 *^#^	0.93 ± 0.88 *
Group 2	10/13	72.65 ± 7.23	75.2 ± 4.1	31.55 ± 7.62	30.22 ± 5.69	26.11 ± 1.76	2.61 ± 1.23	1.36 ± 1.01 ^§^
Group 3	15/11	65.54 ± 6.24	68.7 ± 3.1	28.93 ± 5.07	27.08 ± 4.51	14.10 ± 2.49 ^†‡^	3.23 ± 2.10	1.92 ± 0.81
Group 4	12/14	74.76 ± 8.47	70.7 ± 2.9	30.22 ± 6.48	28.89 ± 5.11	25.67 ± 3.16	3.87 ± 1.71	2.15 ± 1.15

IOP: (mmHg ± SD, standard deviation), intraocular pressure; M: males; F: females. * *p* < 0.001 vs. Groups 3 and 4; ^#^ *p* < 0.05 vs. Group 2; ^§^ *p* < 0.05 vs. Groups 3 and 4; ^†^ *p* < 0.001 vs. Baseline IOP; ^‡^ *p* < 0.01 vs. failure Groups (2 and 4).

**Table 2 jcm-11-03959-t002:** Confocal microscopy parameters.

	Baseline DCD	Baseline GCD	Baseline SMR	Baseline VT	Pre-OP IOP	Pre-OP DCD	Pre-OP GCD	Pre-OP SMR	Pre-OP VT
Group 1	57.57 ± 8.69	28.09 ± 2.82	104.68 ± 8.28	1.72 ± 0.92	28.24 ± 4.57	35.04 ± 7.40 *	30.63 ± 2.78	94.99 ± 11.85 ^#^	1.38 ± 0.82 ^#^
Group 2	59.26 ± 5.04	29.36 ± 4.22	100.68 ± 8.70	1.55 ± 0.73	30.22 ± 5.69	41.98 ± 4.42 ^	28.61 ± 4.07	104.29 ± 9.68 ^¥^	1.33 ± 1.12
Group 3	59.35 ± 9.44	29.52 ± 4.12	103.21 ± 10.65	1.54 ± 0.93	27.08 ± 4.51	55.78 ± 10.65 ^§^	29.15 ± 4.41	96.59 ± 10.57 ^#^	1.15 ± 0.88 ^#^
Group 4	58.27 ± 4.78	28.39 ± 4.79	100.72 ± 9.31	1.55 ± 0.88	28.89 ± 5.11	66.67 ± 4.48 ^§^	24.93 ± 4.43 °	109.11 ± 5.18 ^¥^	1.67 ± 0.71

DCD (cells/mm^2^ ± SD), dendritic cell density; GCD (cells/mm^2^ ± SD), goblet cell density; SMR, stromal meshwork reflectivity (arbitrary grading scale); VT, vascular tortuosity (arbitrary grading scale). * *p* < 0.001 vs. Baseline; ^ *p* < 0.01 vs. Baseline; ^#^ *p* < 0.05 vs. Baseline; ^§^ *p* < 0.001 vs. steroid Groups (1 and 2); ° *p* < 0.001 vs. Groups 1–3; ^¥^ *p* = 0.004 vs. successful Groups (1 and 3).

## Data Availability

Not applicable.

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
