# Peer review of "Topical Steroids and Glaucoma Filtration Surgery Outcomes: An In Vivo Confocal Study of the Conjunctiva"

_jcm, 2022, doi:10.3390/jcm11143959_

Round 1
Reviewer 1 Report
Dear Authors,
I wish to submit my review of the article: "Topical Steroids and Glaucoma Filtration Surgery Outcomes: an In Vivo Confocal Study of the Conjunctiva."
The subject is interesting, and the authors should be commended for their work.
The article is well designed, the findings are novel, and the tables and figures are adequate.
However, some points require proofreading.
1. The English language is sometimes hard to follow. It requires careful proofreading from a native speaker to make it more fluent and check for grammatical and syntax mistakes.
2. Methods: Line 140, page 3: " The cell count software (Heidelberg Engineering GmbH) of the confocal microscope was used to determine the GCD (cells/mm2±SD) in manual mode.". Here, it is not clear both the researchers' number involved in the analysis and the inter-rater reliability if more than one person was involved as you used the "manual mode." (In a nutshell: "The interobserver agreement.").
3. Line 154, page four, see comment 2
Author Response
Reviewer 1
Dear reviewer, I would like to thank you for the time you spent for the revision of the manuscript and for your comments.
- The English language is sometimes hard to follow. It requires careful proofreading from a native speaker to make it more fluent and check for grammatical and syntax mistakes.
As you suggested, a native speaker edited the manuscript.
- Methods: Line 140, page 3: " The cell count software (Heidelberg Engineering GmbH) of the confocal microscope was used to determine the GCD (cells/mm2±SD) inmanual mode.". Here, it is not clear both the researchers' number involved in the analysis and the inter-rater reliability if more than one person was involved as you used the "manual mode." (In a nutshell: "The interobserver agreement.").
- Line 154, page four, see comment 2.
We would like to thank you for the constructive comments. In accordance with your suggestions, we extended paragraphs 2.4, 2.6, and 3.2 by explaining the manual analysis methodology with the interobserver agreement. The interobserver agreement for GCD and DCD was reported in the confocal paragraph of results.

Reviewer 2 Report
The authors reported that pre-treatment of topical steroids in filtering surgery could lead to a less intensive post-operative management. The comments are following.
1) In methods, did all the participants receive the same surgery? For example, was the size and shape of the scleral flap and the number of sutures identical in these surgeries?
2) Did each surgeon make his/her own decision on the timing of additional procedures in this study? Or was there any uniform criteria?
3) How many photographs per case were used to measure each of the ICVM parameters in this study?
4) Was there any relationship between the shape of the bleb (e.g. diffuse bleb or not) and the findings of the IVCM parameters? If so, please describe.
5) Just to confirm, did the authors not examine the postoperative blebs by IVCM? Do they think the IVCM parameters of the bleb conjunctiva correlate with the success rate of the surgery at 1 year postoperatively? Please mention, if they can speculate.

Author Response
Reviewer 2
Dear reviewer, I would like to thank you for the thorough revision of the manuscript and for your insightful comments.
1) In methods, did all the participants receive the same surgery? For example, was the size and shape of the scleral flap and the number of sutures identical in these surgeries?
We thank you for the comments. All patients received the same surgery, MMC-augmented trabeculectomy, as previously reported in lines 102, 103.
We modified the text as follows: “All surgical procedures were performed using a standardized technique, which required the creation of fornix based conjunctival flap, and a 4x4 mm, 300 µm thick scleral flap, which was sutured down with four 10-0 nylon sutures. To limit the post-operative inflammation, the conjunctival flap was finally sutured down using 10-0 nylon sutures” (paragraph 2.2).
2) Did each surgeon make his/her own decision on the timing of additional procedures in this study? Or were there any uniform criteria?
Thank you for the comment. We modified as follows: “Each decision on the timing for additional procedures was evaluated according to post-operative IOP and bleb morphology. If IOP progressively tended to increase and was not considered at target for the patient [19], and bleb features tended to turn into encapsulated or flat, management strategies including laser suture lysis, bleb needling with 5-fluorouracil, or bleb revision with mitomycin-C were sequentially adopted”. We clarified this aspect in the paragraph 2.2.
3) How many photographs per case were used to measure each of the ICVM parameters in this study? We used 15 images per case to measure each of the ICVM parameters, 5 for only portions as described in paragraph 2.6. We better described it in the text.
4) Was there any relationship between the shape of the bleb (e.g. diffuse bleb or not) and the findings of the IVCM parameters? If so, please describe.
We would like to thank you for the insightful comment. We did not consider the bleb shape as an outcome of this study.
In more detail, for the statistical analysis we considered the sample divided into predicted groups based on the surgical outcome, i.e. success or failure. In order to better categorize the subjects enrolled we avoided making a classification based on the morphological bleb features as it would have determined an excessive variability in the data analysis and in the possible correlations with pre-surgical IVCM parameters. In fact, even if adopting standardized bleb classification systems (IBAGS or MBGS), the bleb shape definition may have a certain subjectivity.
However, the literature has shown that pre-operative GCD and a lower DCD are related to a higher incidence of surgical success (with diffuse or cystic bleb shape) or failure in the opposite GCD and DCD combination (flat or encapsulated bleb shape) (Ref. 8,9,30) Further studies could be conducted on a large scale to evaluate how pre-surgical topical steroid therapy can affect bleb morphology after glaucoma filtration surgery, can undoubtedly complete this field or research.
The authors added a comment in the limitations paragraph (section 4).
5) Just to confirm, did the authors not examine the post-operative blebs by IVCM? Do they think the IVCM parameters of the bleb conjunctiva correlate with the success rate of the surgery at 1 year postoperatively? Please mention, if they can speculate.
Dear reviewer the authors confirm that they didn’t perform IVCM after surgery, but your observation is pertinent. In this regard, literature is full of studies that describe the filtration bleb characteristics by using confocal microscopy and that demonstrate how even in the post-operative time, a higher density of goblet cells correlates with an increased rate of surgical success (30) Agnifili et al. described that an increase in GCD on bleb conjunctiva corresponded to a lower inflammatory condition of the entire ocular surface (as steroids determine) and predisposed to surgical success six months after glaucoma filtering surgery, causing positive effects on bleb function. In the same study the authors found lower DCD values and HLA-DR expression in successful compared to failed cases.
Based on these findings we can presume that the higher clinical surgical success rate observed in treated patients, where the conjunctiva appears less inflamed before surgery, may favorably also affect the bleb morphology at cellular level after surgery. Thus, as you correctly hypothesized, it is reasonable retain the confocal bleb parameters and surgical success rate at 1 year may correlate. However, this is a topic of a different study
We discussed this aspect in the limitation section.
